# Phospholipase Family Enzymes in Lung Cancer: Looking for Novel Therapeutic Approaches

**DOI:** 10.3390/cancers15123245

**Published:** 2023-06-19

**Authors:** Sara Salucci, Beatrice Aramini, Anna Bartoletti-Stella, Ilaria Versari, Giovanni Martinelli, William Blalock, Franco Stella, Irene Faenza

**Affiliations:** 1Department of Biomedical and NeuroMotor Sciences (DIBINEM), University of Bologna, 40126 Bologna, Italy; ilaria.versari2@studio.unibo.it; 2Department of Medical and Surgical Sciences (DIMEC), University of Bologna, 40126 Bologna, Italy; beatrice.aramini2@unibo.it (B.A.); anna.bartoletti2@unibo.it (A.B.-S.); franco.stella@unibo.it (F.S.); 3Istituto Scientifico Romagnolo per lo Studio e la Cura dei Tumori (IRST) IRCCS, 47014 Meldola, Italy; giovanni.martinelli2@unibo.it; 4“Luigi Luca Cavalli-Sforza’’ Istituto di Genetica Molecolare, Consiglio Nazionale delle Ricerca (IGM-CNR), 40136 Bologna, Italy; blalockwl@yahoo.com; 5IRCCS, Istituto Ortopedico Rizzoli, 40136 Bologna, Italy

**Keywords:** lung cancer, phospholipase family enzymes, PLC, PLD, PLA

## Abstract

**Simple Summary:**

This review highlights the role of phospholipase in lung cancer, a topic which should be better elucidated considering that phospholipase signaling plays a crucial role in the regulation of multiple cellular processes and its deregulation can promote lung tumorigenesis, favoring lung cancer cell proliferation, migration, and invasion.

**Abstract:**

Lung cancer (LC) is the second most common neoplasm in men and the third most common in women. In the last decade, LC therapies have undergone significant improvements with the advent of immunotherapy. However, the effectiveness of the available treatments remains insufficient due to the presence of therapy-resistant cancer cells. For decades, chemotherapy and radiotherapy have dominated the treatment strategy for LC; however, relapses occur rapidly and result in poor survival. Malignant lung tumors are classified as either small- or non-small-cell lung carcinoma (SCLC and NSCLC). Despite improvements in the treatment of LC in recent decades, the benefits of surgery, radiotherapy, and chemotherapy are limited, although they have improved the prognosis of LC despite the persistent low survival rate due to distant metastasis in the late stage. The identification of novel prognostic molecular markers is crucial to understand the underlying mechanisms of LC initiation and progression. The potential role of phosphatidylinositol in tumor growth and the metastatic process has recently been suggested by some researchers. Phosphatidylinositols are lipid molecules and key players in the inositol signaling pathway that have a pivotal role in cell cycle regulation, proliferation, differentiation, membrane trafficking, and gene expression. In this review, we discuss the current understanding of phosphoinositide-specific phospholipase enzymes and their emerging roles in LC.

## 1. Introduction: Lung Cancer Features

Cancer is a leading cause of mortality and morbidity globally, representing a heavy economic burden on patients, healthcare organizations, and countries in terms of healthcare spending and productivity loss [1,2]. In the United States, cancer is the second most common cause of mortality for men and women after cardiovascular disease, with 609,360 deaths from cancer and 1.9 million new cancer diagnoses recorded in 2022 [3]. The American Cancer Society (ACS) has estimated that 25% of global cancer mortality is due to lung cancer (LC). The highest incidence of LC (53%) is in individuals aged 55 and 74 years, with the incidence dropping to 37% in persons older than 75 years [4]. Cigarette smoking is the most significant risk factor for LC, with active smoking accounting for approximately 85–90% of cases [4]. Other LC risk factors include exposure to radon, occupational exposures, infections, inhalation of pollution particles, and genetic susceptibility [4,5]. However, the multiple factors that may induce LC have not yet been completely elucidated [6].

There are two major histotypes of LC: small-cell lung cancer (SCLC) and non-small-cell lung cancer (NSCLC) [7]. Generally, NSCLC is the most prevalent histotype, accounting for approximately 84% of LC cases, while SCLC accounts for approximately 13% [7,8]. The most common NSCLC histological subtypes are adenocarcinoma (approximately 80% of NSCLCs) and squamous cell carcinoma of the lung (approximately 20% of NSCLCs) [8]. Although research has highlighted the presence of oncogenic driver mutations in adenocarcinoma, common genetic alterations in each histotype have not been identified, and LC remains a disease with high mortality [9]. A key public health challenge is that this type of cancer is generally asymptomatic until the advanced stage, which is associated with a poor prognosis. However, treatment with immunotherapy can improve survival rates [10], with a 5-year overall survival rate of 18.4% following treatment with pembrolizumab/chemotherapy versus 9.7% with chemotherapy alone [11]. This indicates the need to offer more tailored therapies in the treatment of LC [12].

The scientific community has focused its attention on the molecular aspects of cancer. Phospholipases, which are lipolytic enzymes that hydrolyze phospholipid substrates at specific ester bonds, have become a key research area [13]. Phospholipases are widespread in nature and perform diverse functions, from aggression in snake venom to signal transduction, lipid mediator production, and metabolite digestion in humans [14]. The three mammalian families of phospholipases are phospholipases A, C, and D (PLA, PLC, and PLD, respectively). The differences among these groups primarily relate to the type of reaction they catalyze [15]. PLA, PLC, and PLD (Figure 1) are fundamental mediators of intracellular and intercellular signaling, acting as phospholipid-hydrolyzing enzymes that can generate many bioactive lipid mediators, including diacylglycerol (DAG), phosphatidic acid (PA), lysophosphatidic acid (LPA), and arachidonic acid (AA) [14,15,16].

The lipid mediators induced by phospholipases can regulate multiple cellular processes, including tumorigenesis, cancer proliferation, dissemination, invasion, and angiogenesis [17]. Several studies have shown that phospholipase A2 (PLA2) is a crucial regulator of cancer induction or regression through an apoptotic process. In addition, research has identified the role of PLD in cancer development, including in signal transduction, cell proliferation, and anti-apoptotic processes [18]. Specifically, PLD is a downstream transcriptional target of proteins that induce inflammation and carcinogenesis [19]. Therefore, inhibiting the expression or activity of PLD in cancer cells represents an important therapeutic target for inflammation reduction and the sensitization of resistant cancers during chemotherapy [18,19].

Phospholipase C beta (PLCβ) comprises four isozymes (β1–β4) encoded by distinct genes [20]. The PLCβ gene family has been associated with various diseases, including cancer, although the possible connection with NSCLC remains unclear. PLCβ3 is expressed in several tissues [20,21,22]. PLCβ1 is highly expressed in neuronal cells [23], with PLCβ2 in hematopoietic cells [24] and PLCβ4 in specific brain areas [25]. Strassheim et al., (2000) [26] defined PLCβ as a mediator of cancer development for both SCLC and NSCLC. In addition, PLCβ1-augmented expression plays a fundamental role in the activation of neuroendocrine growth factors that induce the progression of SCLC [27]. However, no studies have determined which PLCβ isozymes are connected to cancer and specifically to LC [27].

However, the role of phospholipases in the regulation of tumor cellular processes and their interplay has yet to be elucidated. A thorough understanding of the cancer-associated signaling networks of phospholipases is necessary to determine whether these enzymes can be targeted therapeutically.

## 2. PLC and Lung Cancer

PLCs are present as 13 distinct isozymes, and they can be divided into six sub-groups: PLCβ (1, 2, 3, and 4), PLCγ (1 and 2), PLCδ (1, 3, and 4), PLCε, PLCη (1 and 2), and PLCζ [28]. These enzymes are activated by different stimuli and hydrolyze the highly phosphorylated lipid phosphatidylinositol 4,5-bisphosphate (PIP2), generating two intracellular products, inositol 1,4,5-trisphosphate (InsP3), a calcium (Ca^2+^)-releasing second messenger, and diacylglycerol DAG, which activates protein kinase C [29,30]. PLC isozymes have highly conserved X and Y domains that are necessary for PIP2 hydrolysis. Each PLC contains regulatory domains, including the C2 domain, EF-hand motif, and pleckstrin homology (PH) domain [31]. Each domain has a specific role; for instance, the PH domains bind PtdIns(4,5)P2 or the EF-H domain is involved in supporting guanosine triphosphate (GTP) hydrolysis upon G-protein coupled receptor (GPCR) binding and the C2 domain controls intra- and inter-molecular signaling processes [31,32]. Many studies have highlighted the role of PLC family enzymes in the progression of specific cancer types. These improvements will prompt the discovery of new drugs that target PLC enzymes and will thus pave the way for potential treatments for several diseases for which therapies remain ineffective [31,32].

### 2.1. PLCβ

The C class of phospholipases comprises four isoforms: PLCβ1 (with two spliced variants), PLCβ2, PLCβ3, and PLCβ4 [33,34,35]. PLCβ1 is mainly expressed in the nervous system, namely in the cerebral cortex and hippocampus [36], and in the cardiovascular [37] and musculoskeletal systems [30,38]. PLCβ2 isoforms are present in hematopoietic cells, platelets [39], and breast tissue [40]. PLCβ3 is expressed in the liver, brain, hematopoietic cells, cardiovascular system, and parotid gland, where it mediates proliferation and chemotaxis [39]. PLCβ4 is expressed in the cerebellum and retina for visual processing events after phototransduction [41]. Several studies have demonstrated that the phosphoinositide cycle is present in the cell nucleus, and it is associated with nuclear events, such as mRNA export, DNA repair, and gene transcription in growth, proliferation, cell cycle regulation, and differentiation [42,43,44,45,46]. Alterations of PLC isozyme expression and activation levels are observed in human cancers and are associated with tumor progression. Moreover, PLCβ1 functions as a negative regulator of erythroid differentiation, which was previously reported by in vitro studies on erythroleukemia cells [47]. Due to their backhanded behavior, some phospholipase isozymes have oncogenic roles and others have tumor-suppressive roles based on the environment in which they are located [48]. PLCβ1 was shown to be involved in breast cancer, and there are significant differences in PLCβ1 expression between metastasized and recurrent tumor tissue, highlighting its role in promoting migration in breast cancer [49]. However, few studies have focused on the expression patterns of PLCβ isozymes in various LC subtypes. One study [26] reported that PLCβ1 and PLCβ3 are expressed in SCLC and NSCLC, indicating the role of PLCβ1 in proliferation. In contrast, PLCβ2 and PLCβ4 expression is absent in all SCLC and NSCLC cell lines, and PLCβ1 is more highly expressed in SCLC than in NSCLC. The roles of PLCβ1 in cell viability and proliferation have been studied using the phosphatidylinositol-specific PLC inhibitor ether-linked phospholipid analog (ET-18-OCH3). Strassheim et al., (2000) reported that ET-18-OCH3 inhibited the proliferation of human LC cells [26]. In addition, the increased expression and activity of PLCβ1 mediated by neuropeptide agonists suggests that PLCβ1 can induce the stimulation of neuroendocrine growth factors that support the progression of SCLC. The authors suggest that PLCβ has an important role in the proliferation of both SCLC and NSCLC and that differences in PLCβ1 expression levels may be helpful in the investigation of available diagnostic and therapeutic tools [26]. Recently, Zhang et al., (2019) [27] investigated whether the PLCβ family influenced the prognosis of LC and aimed to identify correlations between NSCLC prognosis and the expression patterns of PLCβ family members. Their findings illustrated the prognostic value of PLCβ members in relation to their mRNA levels in NSCLC. Elevated mRNA expression levels of PLCβ1, PLCβ2, and PLCβ3 were associated with poor overall survival in every NSCLC patient in the study and were significantly associated with poor prognosis in adenocarcinoma cases. In contrast, elevated PLCβ4 mRNA expression was associated with better overall survival in adenocarcinoma patients. Furthermore, PLCβ4 expression is significantly associated with a non-smoking status, while PLCβ1, β2, and β3 expression was associated with both smoking and non-smoking status [27]. Recently, it has been demonstrated that the expression levels of CXCL13 and CXCR5 correlate with LC progression. Authors described that the CXCL13/CXCR5 axis concurred with cell motility in LC cells, which was induced by VCAM-1 expression. Moreover, the PLCβ, protein kinase C alpha (PKCα), and c-Src signaling pathways are involved in CXCL13-promoted cell migration and VCAM-1 expression in LC cells. Pre-treatment with PLCβ, PKCα, and c-Src inhibitors (U73122, GF109203X, and protein phosphatase 2A [PP2]) prevents CXCL13-promoted cell migration and the induction of VCAM-1 expression in LC cells [50].

### 2.2. PLCε

PLCε plays an important role in regulating multiple biological processes that may affect tumor development. Numerous biological functions may be regulated by PLCε; PLCε is involved in inflammation and cancer, while its activity is mainly described with respect to calcium-induced calcium release (CICR) and hypertrophy in the cardiovascular system [35]. Regarding LC, high expression of PLCε was detected in NSCLC tissues and lung cell lines. Both mRNA and protein levels of PLCε in A549 and H1299 cells were higher than those in 293T cells. Moreover, the knockdown of PLCε expression induced apoptosis in NSCLC cells by promoting PTEN methylation and inhibiting the expression of PTEN, which functions as a tumor suppressor gene. In this context, PLCε is described as a novel therapeutic target for NSCLC treatment [51]. Martins et al., (2014) [52] provided novel evidence for the downregulation of PLCε in lung tumors. They observed a reduction in PLCε expression in human lung adenocarcinomas and colorectal and skin tumors and Ras-driven animal tumors, indicating the role of this PLC as a tumor suppressor and suggesting its use as a biological marker in cancer. The same study reported the downregulation of PLCε expression in several NSCLCs. Of note, PLCε expression in these cells can be increased by a histone deacetylase inhibitor (TSA) and DNA methylation inhibitor, indicating the epigenetic control of PLCε regulation in human tumors [52]. In contrast with the results reported by Martins and colleagues, a study conducted by Luo and colleagues described that the mRNA expression of PLCε was higher in NSCLC cells than in non-cancerous cells obtained from adjacent lung tissue from 36 NSCLC patients [53]. In contrast, high levels of PLCε have been shown to decrease the expression of p53 [53]. An analysis of datasets from the Oncomine cancer microarray database indicated low PLCε expression in lung adenocarcinoma tissues compared with normal tissues, confirming a tumor-suppressor role for this protein [54]. Studies on LC describe a novel role for melatonin as an antistemness agent for the treatment of LC, indicating the involvement of the PLC, ERK/p38, β-catenin, and Twist signaling cascades. Using a sphere formation assay, the researchers found that melatonin inhibits sphere formation in two LC cell lines and LC stemness by downregulating the PLC, ERK/p38, and β-catenin signaling pathways [55].

### 2.3. PLC δ

PLCδ has a crucial role in cell signal transduction and cytoskeleton modulation. This isoform is found not only in the plasma membrane but also in the cytoplasm and nucleus, where it is involved in controlling proliferation, differentiation, and cell death [56]. PLCδ1 expression is downregulated in various diseases, such as breast cancer [57] and esophageal squamous cell carcinoma [58], indicating that PLCδ plays an inhibitory role in tumor pathology. In colon cancer, the downregulation of PLCδ1 leads to a reduction in cellular apoptosis and an increase in cellular proliferation, invasion, and migration; therefore, PLCδ1 is described as a potential biomarker of colon cancer [59]. PLCδ1 is involved in energy metabolism, Ca^2+^ homeostasis, and intracellular motility. Recently, it has been demonstrated that PLCδ1 expression is downregulated in lung adenocarcinoma cells. In A549 cells, PLCδ1 overexpression inhibits proliferation, cell invasion, migration, and epithelial–mesenchymal transition in LC cells. Moreover, in A549 cells, ORC1 binds to the PLCδ1 promoter, significantly reducing PLCδ1 expression, which may affect the progression of LC [60]. PLCδ1 acts as a tumor-suppressor gene in various tumors, and it is epigenetically silenced through hypermethylation in esophageal squamous cell carcinoma, breast cancer, gastric cancer, and chronic myeloid leukemia [32].

### 2.4. PLC γ

PLCγ is ubiquitously expressed and exists in two isoforms, PLCγ 1 and PLCγ2, each with distinct functions in a variety of cell types and disease states. Both PLCγ isoforms are mutated in many cancers, and PLCγ1 positively drives tumorigenesis in the lung [28]. Evidence indicates a key role for PLCγ 1 in lung metastasis formation in that the blockade of PLCγ 1 reverts metastasis formation and may thus have potential therapeutic applications in the clinical treatment of tumor metastasis. The downregulation of PLCγ results in an almost complete inhibition of breast cancer-derived lung metastasis formation [61]. PLCγ1 plays a crucial role in the proliferation and migration of tumor cells process. Some studies have described a link between PLCγ and autophagy in tumor progression [62]. Recently, Timsah et al., (2015) [63] reported that PLCγ1 expression is involved in tumor formation and recurrence-free survival in 150 lung adenocarcinoma patients. High levels of fibroblast growth factor receptor 2 (FGFR2) and PLCγ1 expression of FGFR2 and low levels of Grb2 expression are consistent with poor prognosis; in contrast, low levels of FGFR2 and PLCγ1 expression in the presence of high levels of Grb2 expression correlate with good prognosis. In NSCLC, PLCγ1 is involved in the regulatory mechanism of programmed death ligand 1 (PDL1), which increases the activation of PLCγ1 by the epidermal growth factor receptor (EGFR). Programmed death 1 (PD-1), a checkpoint protein inhibitor, and its ligand PD-L1 are targets of modern cancer immunotherapy [64]. The binding of PLCγ1 to a conserved segment of PD-L1 leads to increased phosphorylation and catalytic stimulation of PLCγ1 after the stimulation of cells with EGF. Moreover, when activated, PLCγ1 triggers Ca^2+^ flux and activates Rho GTPases and PKC, which promotes an aggressive phenotype; an anti-PD-L1 antibody can prevent these constitutional functions of PD-L1. In this model, PD-L1 can amplify EGFR signaling, which improves our understanding of the resistance of EGFR tumors to immunotherapy [65]. Furthermore, a recent study reported that PLCγ1 is a critical mediator of FGFR1 signaling in SCLC. The study suggests that PLCγ1 is involved in FGFR1 signaling to favor tumorigenic progression in precancerous neuroendocrine cells and is required for SCLC development in vivo, independently of ERK1/2, AKT1, or STAT1. Data suggest that PLCγ1 activation by FGFR1 cooperates with transcriptomic changes related to neural differentiation, indicating a role for PLCγ1 in regulating neuroendocrine differentiation in SCLC. Therefore, these data suggest that PLCγ1 is a potential therapeutic target in SCLC and add SCLC to the growing list of malignancies in which PLCγ1 activity is implicated [66]. Investigations have also focused on the pro-oncogenic role of PLCγ1 in LC. EphA2A is a receptor tyrosine kinase (RTK) identified as a potential driver of NSCLC as it is commonly overexpressed in NSCLC and associated with poor clinical outcomes [67]. Using human LC cell lines, Song et al., (2020) [68] demonstrated that EphA2 phosphorylates PLCγ1 and promotes aberrant cell proliferation. The downregulation of EphA2 activity reduces PLCγ1 phosphorylation and suppresses lung carcinoma cell growth in vitro. EphA2 is highly expressed in many KRAS-mutant LC cells and has been shown to regulate tumor malignancy. PLCγ1 promotes tumor cell proliferation in KRAS-mutant LC cells, suggesting that PLCγ1 is a novel interactor of the EphA2 RTK in LC cells and that the EphA2–PLCγ1 signaling axis could be a promising therapeutic strategy for treating LC [68]. Another study highlighted a mechanism of cancer cell adaptation to hypoxia with potential therapeutic value for PLCγ1. Lipidomic screening was used to discover that PLCγ1 is suppressed during hypoxia in KRAS-mutant human lung adenocarcinoma cancer cell lines. In this cellular model, the downregulation of PLCγ1 in hypoxia decreased oxidative cancer cell metabolism, prevented the formation of mitochondrial reactive oxygen species, and drove tumor bioenergetics toward glycolysis by preventing Ca^2+^ entry into the mitochondria.

Moreover, using a genetically engineered mouse model of lung adenocarcinoma development driven by KrasG12D (the KrasLSL-G12D mouse) and several biochemical and genetic approaches, including in vivo xenotransplantation studies, researchers demonstrated that a loss of function of PLCγ1 leads to the increased expression of glycolytic genes, increased tumor growth, and decreased survival. In patients with KRAS-mutated lung adenocarcinomas, the decreased expression of PLCγ1 corresponds to increased expression of hypoxia markers and poor patient survival [69].

### 2.5. PLC Concluding Remarks

As discussed, PLC activation is followed by the generation of two second messengers, DAG and IP3, which induce the endoplasmic release of Ca^2+^ with the activation of PKC. PKC phosphorylates downstream effectors, such as AKT, which control several cellular functions [70,71]. Consequently, the dysregulated metabolism of phosphoinositide is associated with several diseases, including cancer. PLC plays a significant role in the prognosis of LC. Increased mRNA expression of PLCβ1–3 has been evaluated as a biomarker of poor survival in LC, and increased expression of PLCε and PLCγ can be considered parameters for the diagnosis of NSCLC. In contrast, increased mRNA levels of PLCβ4 are associated with a favorable prognosis in lung adenocarcinoma. Finally, only PLCδ is downregulated in lung adenocarcinoma, and its upregulation inhibits LC cell proliferation and invasion. Therefore, the PLC family of enzymes plays a crucial role in LC, and their expression levels have significant value in the prognosis and progression of the disease.

## 3. PLD and Lung Cancer

PLD hydrolyzes phosphatidylcholine to produce choline and PA, which is an essential regulator of mammalian target of rapamycin (mTOR) stability. PLD controls multiple biological functions; for example, it is involved in fatty acid synthesis, cytoskeleton dynamics, and secondary messenger transduction. It has an essential role in the regulation of tumor cell proliferation and metastasis [72] as it is involved in cancer cell apoptosis inhibition [73]. When PLD is overexpressed, it induces the activation of two oncogenic targets, mitogen-activated protein kinase (MAPK) and mTOR, and inhibits tumor suppressor p53 (TP53) and PP2 [74]. Increased PLD activity has been documented in various tumors, including human gastric, colorectal, breast, renal, prostatic, and papillary thyroid cancers [75,76,77]. PLD expression is regulated by a complex network of enzymes and substrates, including PKC, GTP-binding proteins, and Ca^2+^. Two PLD isoforms, PLD1 and PLD2, have different regulatory and functional roles in most mammalian tissues. Human PLD1 and PLD2 show a similar structure with conserved tandem PX and PH domains, two HKD catalytic domains that combine to form a single active site, and a conserved C-terminus [78]. Between the two catalytic HKD domains, there is a flexible loop of 140 amino acids, which distinguishes PLD1 from PLD2. PLD1 is localized in the cytoplasm and can be activated by PKC and the small GTPases ARF and Rho. PLD2, which has high basal activity compared with PLD1, is localized on the plasma membrane and is activated by tyrosine kinases. PLD2 activity is negatively regulated by cellular targets localized in the cytosol or on the cell membrane, such as α-synuclein and α-actinin, or positively modulated by signaling pathways and molecules that also control PLD1 activity [79]. Furthermore, PA, a product of PLD, contributes to the activation of several regulatory proteins, such as Raf-1 kinase and PLC, which leads to mTOR activation [80]. In numerous human cancers, both PLD isoforms are aberrantly increased, and recent studies have suggested that PLD is a downstream transcriptional target of proteins, such as Sp1 and NFκB, which lead to inflammation and carcinogenesis [81]. In LC, PLD1 plays a significant role in maintaining PA production and enhancing downstream signaling when PLD2 is downregulated. PLD1 interacts with aldolase, a metabolic enzyme that inhibits PLD2; this inhibition is compensated by PLD1 activity, which enhances lung carcinogenic features, such as proliferation, repair, and anti-apoptotic functions [82]. In vitro studies on LC cells indicate that the aberrant expression of PLD inhibits apoptotic cell death, while its downregulation stimulates apoptosis and autophagy in H460 human LC cells with the involvement of noncoding RNA, a known player in cancer [83]. Furthermore, the activation of PLD1 by bradykinin and sphingosine 1 is involved in the PKC signaling pathway in A549 human lung adenocarcinoma cells, indicating an association between PLD1 and NSCLC [80]. Therefore, PLD upregulation and mTOR pathway activation represent two conditions that favor LC growth, and their activity is strictly dependent on PKC overexpression, which enhances the proliferation and migration of lung tumor cells. In this scenario, PKC acts as a driving factor, and PLD is a crucial regulator of PKC-dependent mTOR activation; as a result, cancer cells become more susceptible to PLD and mTOR inhibitors [84]. Another crucial player in cancer is interleukin-8 (IL-8), a proinflammatory CXC chemokine that is considered a regulatory factor in the tumor microenvironment, namely when it is overexpressed [85]. IL-8 signaling promotes the activation of PI3K, PLC, and PLD, followed by the activation of AKT and PKC, which in turn activates ERK1/2 signaling. IL-8 levels are high in LC cell lines and in the sera of LC patients with bone metastases [86]. Blocking the PKC/ERK1/2 or PLD/AKT pathways using specific inhibitors causes a reduction in bone metastases in LC patients [87]. Furthermore, it has been demonstrated that PLD plays an interesting role in the control of the tumor microenvironment. PLD1 inhibition affects tumor neovascularization and growth through AKT and MAPK downregulation, resulting in reduced tumor angiogenesis. This has been evidenced in mice deficient in PLD1 that exhibit a reduced risk of lung metastasis [88]. Pazhouhandeh and colleagues demonstrated PLD involvement in NSCLC and cancer in smokers. In both conditions, PLD activated signaling pathways, such as EGF/EGFR, which are known players in the initiation and progression of several tumors [89]. Furthermore, the same researchers demonstrated the involvement of PA in NSCLC. In NSCLC patients, PA, the product of PLD enzymatic activity, can be converted to LPA, which controls various cancer-related processes, such as proliferation, growth, and survival [90]. Finally, it has been demonstrated that the increased risk of LC is strictly associated with gene polymorphisms, specifically for genes that encode the enzymes involved in the metabolic activation and detoxification of tobacco carcinogens to repair DNA damage. PLD also falls within this genetic susceptibility. It has been demonstrated that variations in PLD1 DNA sequences lead to aberrant PLD activity, with consequent individual differences in LC susceptibility. Therefore, PLD1 polymorphisms also play a role in the development of NSCLC and can be considered an important marker of genetic susceptibility to LC [91].

### PLD Concluding Remarks

The phospholipase D signaling pathway is involved in lung cancer, and it is phosphorylated by PKC, a downstream component of the PLC pathway as well. PLD activity increases in response to mitogenic signals (e.g., FGF) and oncogene expression. PLD activates RAS [92], promoting cell proliferation and transformation, and through its mediators, it is involved in the activation of RAS–MAPK and mTOR signaling. In general, the overexpression of PLD in LC promotes cell proliferation and apoptosis inhibition through activation of the ERK pathway. In contrast, the inhibition of PLD induces apoptosis and autophagy in human LC cells [84].

## 4. PLA and Lung Cancer

PLA is a large family of esterases that catalyze the hydrolysis of fatty acid ester bonds in phospholipids to generate fatty acids and lysophospholipids. Lysophospholipids and their metabolites are signaling molecules that are involved in cellular adaptation to stress, developmental processes, and various diseases in both plants and animals [93]. Current evidence suggests that PLA represents a crucial target involved in the modulation of intracellular molecular pathways, leading to the generation of lipid mediators, which control and regulate cell proliferation, survival, differentiation, motility, and tissue vascularization [94]. Therefore, the deregulation of PLA is involved in the pathogenesis of several diseases and cancer development and progression. The PLA family includes two enzymes, PLA1 and PLA2, which are involved in removing fatty acids from the glycerol structure of target phospholipids. PLA1 and PLA2 catalyze the hydrolysis of the ester bond at the sn-1 or sn-2 position of phospholipids, respectively [95]. PLA1 enzymes are currently not well understood, and their crystal structures and biological functions remain mostly unknown. PLA1 is involved in the turnover and remodeling of cellular glycerophospholipids, and some PLA1s play a specific role in the production of 2-acyl-1-lysophospholipids, which are lysophospholipid mediators [96]. The literature reports PLA1 involvement in the production of lysophospholipids, which are implicated in angiogenesis and protein transport [97] in cardiovascular diseases and atherosclerosis pathology. In contrast, PLA2 family enzymes have been widely investigated, and their involvement in the regulation of some biological processes, such as inflammation, the immune system, and cancer, was demonstrated in both in vivo and in vitro studies [98]. PLA2s contain seven highly conserved disulfide bridges. The binding of calcium ions is provided by the conserved aspartate residue, carbonyl oxygen in the tyrosine residue, and glycines in the calcium-binding loop. The activity of most PLA2 enzymes depends on their ability to interact with lipid aggregates [99]. Two main products originate from the PLA2 catalytic process. Moreover, the presence of polyunsaturated fatty acids (mostly AA), lysophospholipids, and the overproduction of these lipid mediators lead to inflammation and tissue degeneration [100]. Therefore, PLAs are essential for controlling homeostasis and the pathogenesis of various inflammatory conditions, including rheumatoid arthritis, atherosclerosis, multiple forms of cancer, and cardiovascular disease [101]. Polyunsaturated fatty acids are involved in maintenance of the phospholipid plasma membrane composition, generating epidermal lipid barriers, and energy production in the β-oxidation metabolic pathway [102,103]. Among polyunsaturated fatty acids, AA is a crucial substrate for lipoxygenases (LOXs), cyclooxygenases (COXs), and cytochrome p450 enzymes, which function to produce eicosanoids [104]. Eicosanoids are strong inflammatory molecules that are crucial in the establishment of a highly inflammatory tumor microenvironment, which is a hallmark of cancer [105]. In contrast, lysophospholipids function as extracellular mediators involved in the control of cellular Ca^2+^ homeostasis, proliferation, survival, migration, and adhesion. Lysophospholipids generate several cellular metabolites, including LPA, which can bind to G protein-coupled receptors, and the deregulation of LPA effectively stimulates cancer cell proliferation and migration [106,107,108]. Furthermore, lysophospholipids are precursors of platelet-activating factor (PAF), which is a potent inflammatory lipid mediator [109]. PLA2s are divided into six subfamilies; among these, secreted PLA2s (sPLA2s), cytosolic PLA2s (cPLA2s), and Ca^2+^-independent PLA2s (iPLA2s) are involved in inflammatory and oncological diseases [110].

### 4.1. sPLA2

The sPLA2 group was the first type of PLA2 to be discovered. These enzymes are found in animal venoms, synovial fluid, and various mammalian tissues. sPLA2s are classified into 18 main groups and have been described as carcinogenic mediators due to the metabolic activity of their reaction products, which include eicosanoids. sPLA2 plays a pro-tumorigenic role in breast, lung, ovarian [111,112,113], and other cancers [114]. sPLA2 groups IIa and X are highly expressed in tumors and exhibit significant enzymatic activity related to cancer development and progression. Among the sPLA2 subtypes, sPLA2-IIa is the most studied and is closely associated with the progression of many types of cancers, including breast, prostate, esophageal adenocarcinoma, and LC [115,116,117]. sPLA2-IIa induces the synthesis of prostaglandins and release of AA, which leads to the activation of inflammatory molecules, such as NF-κB, COXs, LOXs, and PGE2, which are involved in cancer progression. In addition, sPLA-IIa is involved in modulating tumor progression via metastasis, apoptosis, and inflammatory processes [118,119]. Regarding LC, sPLA2-IIa can induce NF-κB activation in a dose- and time-dependent manner through the phosphorylation of HER2 and HER3. Therefore, sPLA2-IIa appears to be a potential substrate for EGFRs and stimulates HER/HER2–PI3K–Akt–NF-κB signaling, the activation of which promotes LC development and progression [120]. In addition, several studies have reported an association between high plasma levels of sPLA2-IIa and poor prognosis in certain cancers [121]. This can be explained by its relationship with NF-κB, which is involved in inflammatory and innate immune responses and promotes cell proliferation and migration by suppressing apoptosis [120,122]. Furthermore, sPLA2-IIa, in addition to promoting AA release, produces several lipid products, including lysophosphatidylcholine and free fatty acids, which induce the phosphorylation of PKC by activating cPLA2 and the MAPK pathway. This generates a positive feedback loop, as cPLA2 activation can produce AAs and other lipids that are responsible for sPLA2 activation [123,124]. In this scenario, the modulation of sPLA2-IIa expression plays a key role in downregulating the expression of NF-κB in NSCLC [119]. Zhang et al., (2022) demonstrated that emodin, a natural compound with various pharmacological effects, which is found in the roots and bark of several oriental herbs, can inhibit the proliferation of KRAS-mutant lung cell lines by decreasing the activation of sPLA2-IIa and NF-κB pathways. Furthermore, another study reported that sPLA2-IIa inhibition decreased the production of PGE2 (which contributes to LC invasiveness) and the proliferation of lung adenocarcinoma cells [125]. These results suggest that sPLA2-IIa could represent a potential therapeutic target against LC, and there is evidence for utilizing sPLA2-IIa expression as a biomarker for the prediction of the clinical outcomes of LC patients [114,126]. Therefore, sPLA2-IIA upregulation in the lung favors tumorigenesis, proliferation, cancer cell survival, and increased local inflammation [18].

### 4.2. cPLA2

cPLA2 is widely distributed in most human tissues and is involved in allergic responses and inflammatory damage associated with colorectal, gastrointestinal, prostate, and lung carcinomas [127,128,129]. Its expression is correlated with the upregulation of COX-1 and COX-2, which are molecules involved in inflammation that increase the risk of tumor development [130]. After anti-cancer treatment, the expression of COXs and cPLA2 significantly decreases, suggesting a high correlation between their expression levels and LC treatment status. Xin and colleagues [129] demonstrated that cPLA2 expression is significantly increased in LC patients, and its expression was reduced after anticancer treatment; this suggests its important role as a biomarker in the early detection of LC and as a crucial enzyme in lung tumor development, with involvement in driving tumor migration and invasion [131]. In addition, cPLA2 and COX-2 upregulation associated with high levels of prostaglandin production was reported in NSCLC patients [132], indicating that the cPLA2–AA–COX-2 pathway could be crucial and strategic for LC diagnosis and treatment. In this regard, Xin and colleagues (2019) demonstrated that cPLA2, COX-2, COX-1, mPGES, PGE2, and PGI2 are significantly increased in LC patients independently from the histological subtype, and their levels decreased after treatment, suggesting a strict association between these molecular targets and LC [129]. Several substrates can control cPLA activity, including MYC. The oncolytic activation of MYC leads to lung adenocarcinoma progression. Moreover, lung tumor tissue exhibits increased levels of signaling precursor phospholipids, namely free AA, released from membrane phospholipids by cPLA2. In tumors, high levels of cPLA2 are linked to MYC upregulation. In addition, high expression of cPLA2 was observed in mitotic cells, suggesting a primary role for cPLA2 or its metabolites in proliferation. cPLA2 can be activated by phosphorylation via the MAPK pathway, and its effectors include tumor necrosis factor α and IL-1β, which strictly correlate with MYC activity [133].

### 4.3. iPLA2

iPLA2s are enzymes that do not require Ca^2+^ for their catalytic activity, and they are thus designated as Ca^2+^-independent. iPLA2s are ubiquitously expressed and control a variety of biological processes, including fat catabolism, cell differentiation, the maintenance of mitochondrial homeostasis, phospholipid remodeling, cell proliferation, signal transduction, and cell death. Consequently, their dysregulation is critical in the development of many diseases [134]. The most extensively studied enzymatic isoforms of iPLA2 are the membrane-associated iPLA2γ and the cytosol-associated iPLA2β. Some studies suggest that iPLA2β promotes cancer cell growth through transduction pathways that involve EGFRs, MAPKs, the E3 ubiquitin-protein ligase mdm2, the tumor suppressor protein p53, and the cell cycle regulator p21 [92,135]. Sparse data are available in the literature regarding the involvement of iPLA2s in LC. Some studies report that iPLA2 overexpression is correlated with cancer cell proliferation in both in vitro and in vivo models [136,137]. iPLA2 signaling has mostly been studied in correlation with peroxiredoxins, a family of thiol-dependent peroxidases (PRDXs) that promote tumorigenesis in certain cancer types. PRDX6 is crucial for the maintenance of lung homeostasis as it stimulates antioxidant defense and correct signaling. However, its deregulation, correlated with high expression of iPLA, is associated with various lung pathologies and diseases [138,139]. This relationship is supported by research that has demonstrated that iPLA2 downregulation reduces lung inflammation and oxidative stress [134], thereby preventing cancer development.

### 4.4. PLA Concluding Remarks

PLA2 isoform expression is associated with tumor progression. The activation of ERK and the presence of cytoplasmic Ca^2+^ leads to the phosphorylation of cPLA2, while iPLA2 is phosphorylated by PKC. When iPLA2 and cPLA2 are overexpressed, they hydrolyze several phospholipids to generate the final products, including prostaglandins and leukotrienes, respectively, which are involved in COX and LOX pathway activation to induce inflammation and tumorigenesis. Furthermore, cPLA2 can be activated by sPLA2 as well, which plays a critical role in LC progression. sPLA2 upregulation stimulates NF-κB activity, contributing to LC growth and invasion [113], while its inhibition induces LC cell apoptosis, as demonstrated by in vivo and in vitro studies [93,119,140,141].

## 5. Conclusions and Future Perspectives

As described above, PLC, PLA, and PLD are essential molecules that play critical roles in the control of intracellular and intercellular signaling. They act as phospholipid-hydrolyzing enzymes and produce bioactive lipid mediators (DAG, LPA, PA, and AA) that control and regulate multiple cellular processes, including proliferation, migration, invasion, and angiogenesis [48]. Phospholipases can be activated by multiple extracellular signals, such as hormones, growth factors, and lipids, which bind or activate RTKs or G-protein-coupled receptors [48]. Therefore, phospholipases are necessary mediators of cellular signaling and are essential for the maintenance of normal cellular physiological functions; furthermore, their deregulation is involved in tumorigenesis. In most cases, phospholipase expression leads to the activation of cellular signals [47], which are also referred to as survival signals (e.g., mTOR), which suppress apoptotic cell programming and thereby play a relevant role in tumor inhibition. However, cancer is heterogeneous and characterized by a complex microenvironment, and PLC, PLD, and PLA appear to function as arbitrators between growth promotion and inhibition in cancer cells.

Data collected demonstrate the critical involvement of these bioactive lipid mediators as essential players in LC, as schematized in Figure 2 and Figure 3.

It is essential to motivate the scientific community to investigate and better understand the molecular mechanisms involving the phospholipase family of enzymes as pathological mediators in LC. Furthermore, these enzymes could function as potential biomarkers or anticancer targets for LC therapy. Future studies are needed to establish how phospholipase enzymes regulate cancer-associated cellular processes and their nature and their relative impact on cell motility and cell proliferation in specific LC types.

## Figures and Tables

**Figure 1 cancers-15-03245-f001:**
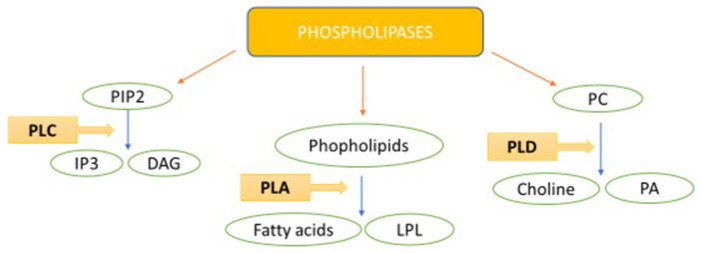
The main products catalyzed by PLC, PLA, and PLD. Phosphatidylinositol 4,5-bisphosphate (PIP2), inositol trisphosphate (IP3), diacylglycerol (DAG), lysophospholipid (LPL), phosphatidylcholine (PC), phosphatidic acid (PA).

**Figure 2 cancers-15-03245-f002:**
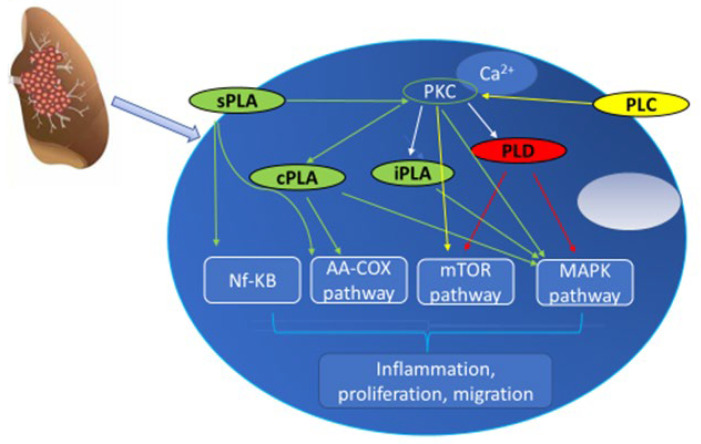
Phospholipase deregulation in lung cancer promotes inflammation, proliferation, and migration, involving different signaling pathways.

**Figure 3 cancers-15-03245-f003:**
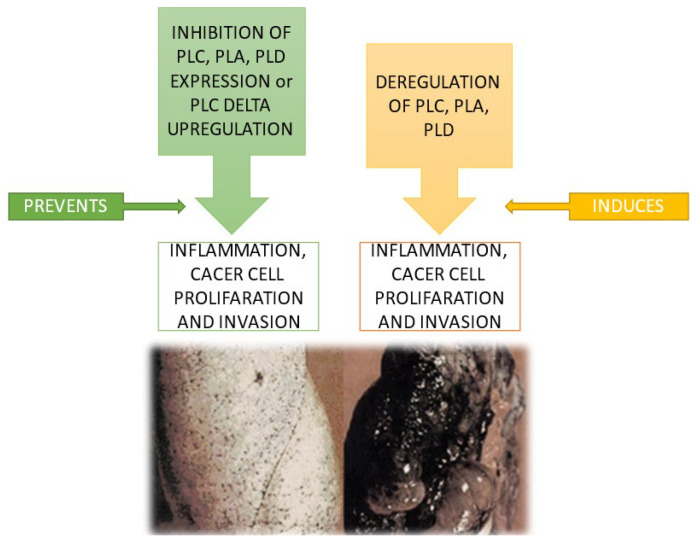
Involvement of PLC, PLD, and PLA in LC.

## Data Availability

Not applicable.

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
