# Peer review of "Phospholipase Family Enzymes in Lung Cancer: Looking for Novel Therapeutic Approaches"

_cancers, 2023, doi:10.3390/cancers15123245_

Round 1
Reviewer 1 Report
Authors aimed to review current understanding of phosphoinositide-specific phospho-lipase enzymes and their emerging roles in lung cancer. Readers would benefit from an extensive and conclusive review article from this perspective. Interestingly, authors formatted the manuscript as a research article that led to a disconnection of each session of three groups of phospholipases and the contents in discussion. Hence, I have following main and minor suggestions on the manuscript.
Main points:
1. Readers would benefit from a general introduction on molecular composition and activation mechanism of phospholipases. In addition, a brief introduction of the known activating mechanism of three main groups of phospholipases in long cancer would be appreciated.
2. The session of Materials and Methods are not necessary for such a review manuscript. Discussion should be included in the appropriate section of manuscript.
3. The writing of manuscript is often not ambiguous and easy to be misinterpreted or confusing. Throughout the manuscript, essential references are needed, and paragraphs need conclusive sentences.
Minor points:
1. Authors mentioned four major families of phospholipids and only three types listed. A brief explanation is needed here.
2. Line 93-102, is there a particular reason for such a disordered explanation of PLCb isoforms?
3. The following expression is confusing--- “The C class of phospholipases comprises four isoforms with spliced variants: PLCβ1, 133 PLCβ2, PLCβ3, and PLCβ4 [32-34].”
4. Frequent errors and typos in the manuscript, such as “Both PLCγ isoforms are mutated in many cancers, and PLCγ1 in particular positively drives 230 tumorigenesis ofin the lung [27]. Evidence indicates a key role for PLCγ 1PLCγ1 in lung 231 metastasis formation in that, as the blockade of PLCγ 1PLCγ1 reverts metastasis formation and, therefore, could may thus have potential therapeutic applications forin the clinical treatment of tumor metastasis.”
The manuscript will benefit from a thorough copyediting.
Author Response
Dear Editor,
Thanks to the useful suggestions of the Reviewers. Here you can find the revised manuscript and the answers to Reviewer’s requests.
We would like to underline that we have added a new author dr William Blalock, our close collaborator, for the language editing and supervision of the manuscript.
Reviewer 1
Authors aimed to review current understanding of phosphoinositide-specific phospho-lipase enzymes and their emerging roles in lung cancer. Readers would benefit from an extensive and conclusive review article from this perspective. Interestingly, authors formatted the manuscript as a research article that led to a disconnection of each session of three groups of phospholipases and the contents in discussion. Hence, I have following main and minor suggestions on the manuscript.
Main points:
- Readers would benefit from a general introduction on molecular composition and activation mechanism of phospholipases. In addition, a brief introduction of the known activating mechanism of three main groups of phospholipases in long cancer would be appreciated.
The structural characterization and activation of each phospholipase group has been added as kindly suggested by the Reviewer.
- The session of Materials and Methods are not necessary for such a review manuscript. Discussion should be included in the appropriate section of manuscript.
According to the Reviewer, the session of Materials and Methods has been deleted.
- The writing of manuscript is often not ambiguous and easy to be misinterpreted or confusing. Throughout the manuscript, essential references are needed, and paragraphs need conclusive sentences.
Conclusive sentences have been added as “Concluding remarks” for each phospholipase group and new references have been updated.
Minor points:
- Authors mentioned four major families of phospholipids and only three types listed. A brief explanation is needed here.
The authors have been corrected “four” with ”three mammalian”.
- Line 93-102, is there a particular reason for such a disordered explanation of PLCb isoforms?In line 93-102 the sentence has been rewritten for a better understanding.
- The following expression is confusing--- “The C class of phospholipases comprises four isoforms with spliced variants: PLCβ1, 133 PLCβ2, PLCβ3, and PLCβ4 [32-34].” The sentence has been corrected
The sentence has been corrected.
- Frequent errors and typos in the manuscript, such as “Both PLCγ isoforms are mutated in many cancers, and PLCγ1 in particular positively drives 230 tumorigenesis ofi n the lung [27]. Evidence indicates a key role for PLCγ 1PLCγ1 in lung 231 metastasis formation in that, as the blockade of PLCγ 1PLCγ1 reverts metastasis formation and, therefore, could may thus have potential therapeutic applications forin the clinical treatment of tumor metastasis.”
In the section relative to PLCγ, the frequent editing errors have been corrected as suggested by both the Reviewers.
Reviewer 2 Report
This is a comprehensive review of the role of phospholipase gene family members in lung cancer. PL is an under-appreciated protein family in cancer development and it is appropriate to review this topic and examine the latest progress made. However, section 3.4 needs editing due to some unexplained digital/word processing errors scattered throughout.
Main points:
Reference 10 is related to quality of life issues rather than focusing on immunotherapy outcomes in LCs.
It is not clear the rationale for choosing this class of enzymes as a major focusing this review and why LCs. The role of phospholipases and their associated signaling events in cancer, particularly related to PKC, have been exhaustively studied in the 80s and 90s by Bernard Weinstein. But the field then switched its focus to PTEN, the lipid phosphatase for PIP3 in late 90s. The “lipid” field in cancer research pretty much dominated by PTEN and PI3-kinase signaling in the subsequent 26 years.
Overall, the authors provided a structured and exhaustive review of different PL isoforms in different LC subtypes. There are also good descriptions of various underlying mechanisms of how PLs promote or attenuate carcinogenesis.
Minor points:
The authors may break up some of the writing into shorter paragraphs instead of having a long block of texts.
Line 234 edit: “down-regulationDownregulation”
Line 236 edit “hasplays” should be “has played”
Line 237 edit “described the turn-out of a link” should be “described a link”
Edit multiple errors for PLCg1 naming in section 3.4. which is very strange.
Multiple very strange errors which are likely to be software-related such as PLCg1 itselfPLCg1 [line 251], showeddemonstrated [line 270], phosphorylatedphosphorylates [line270], promotedpromotes [line 271], lessdecreased [line 283], identifiedresearchers [line 288].
Please edit section 3.4 for multiple typos.
Author Response
Dear Editor,
Thanks to the useful suggestions of the Reviewers. Here you can find the revised manuscript and the answers to Reviewer’s requests.
We would like to underline that we have added a new author dr William Blalock, our close collaborator, for the language editing and supervision of the manuscript.
This is a comprehensive review of the role of phospholipase gene family members in lung cancer. PL is an under-appreciated protein family in cancer development and it is appropriate to review this topic and examine the latest progress made. However, section 3.4 needs editing due to some unexplained digital/word processing errors scattered throughout.
Main points:
- As suggested by the Reviewer, the correct reference on lung immunotherapy has been added.
Dong et al., 2019
- It is not clear the rationale for choosing this class of enzymes as a major focusing this review and why LCs. The role of phospholipases and their associated signaling events in cancer, particularly related to PKC, have been exhaustively studied in the 80s and 90s by Bernard Weinstein. But the field then switched its focus to PTEN, the lipid phosphatase for PIP3 in late 90s. The “lipid” field in cancer research pretty much dominated by PTEN and PI3-kinase signaling in the subsequent 26 years.
Our research group, mainly prof Irane Faenza, has been studied plc signaling for years and she demonstrated the importance of these enzymes in proliferation, differentiation and carcinogenesis. Our group start studying recently lung cancer mechanisms thanks to collaboration with thoracic surgeons. We have had very encouraging data, so we thought to advance this topic for the special issue “Prognostic Biomarkers of Lung Cancer”.
Minor points:
- Some sentences have been corrected into shorter paragraphs.
- In the section relative to PLCγ, the frequent editing errors have been corrected as suggested by both the Reviewers.
Round 2
Reviewer 1 Report
Authors have addressed the suggestion adequately.